# Estimating Antarctic surface melt rates using passive microwave data calibrated with weather station observations.

Valeria Di Biase[1], Peter Kuipers Munneke[1], Bert Wouters[2], Michiel R. van den Broeke[1], and Maurice van Tiggelen[1]

[1]Institute for Marine and Atmospheric Research, Department of Physics, Utrecht University, Utrecht, the Netherlands
[2]Department of Geoscience & Remote Sensing, Delft University of Technology, Delft, the Netherlands

**Correspondence:** Valeria Di Biase (`vdibiase@edf.org`)

**Abstract.**

We present a dataset of Antarctic annual surface melt rates (6.25 km resolution, 2011-2021) from 19 GHz Special Sensor Microwave Imager/Sounder (SSMIS). First, melt occurrence is detected via thresholds for brightness temperature, diurnal variation, and winter anomaly, calibrated with Automatic Weather Station (AWS) data. Second, AWS-driven surface energy balance modeling yields an empirical relation between annual melt days and water-equivalent melt volume. SSMIS-derived melt volumes correlate well with AWS-based melt estimates ($R^2 = 0.83$). Compared to QuikSCAT and RACMO2.4p1 outputs, SSMIS captures a similar spatial melt pattern but estimates a total melt volume approximately 15% lower than RACMO2.4, on the decadal average.

## 1 Introduction

The occurrence of surface melt on the Antarctic Ice Sheet constitutes a key indicator of cryospheric change, with profound implications for ice-shelf stability, glacier dynamics, and continental mass balance. Therefore, it is essential to make observations of surface melt, both to monitor change, and to collect data for the evaluation and development of models.

Surface melt occurs from a surplus of energy in the surface energy budget (Van Den Broeke et al., 2004). If the balance of radiative and turbulent energy fluxes is positive, and the surface is at the melting point, the excess energy is used for melting of the surface snow or ice. For snow, the surface albedo is a dominant driver of the energy budget. Because snow albedo is high, a small albedo change leads to large changes in the available surface energy.

In Antarctica, most surface melt percolates into the firn layer, and refreezes, rather than running off into the ocean (Van Wessem et al., 2018). Although the direct contribution of meltwater runoff to the negative Antarctic mass balance is very small, the indirect effect of surface melt on ice-sheet mass balance is important. In the Antarctic Peninsula, recent warming has increased surface melt (Cape et al., 2015). Refreezing meltwater has depleted firn air (Holland et al., 2011) and promoted the formation of meltwater ponds, a precursor for hydrofracturing (Scambos et al., 2000; Kuipers Munneke et al., 2014). A link with the sudden collapse of the Larsen A and B ice shelves is thereby implied (Dunmire et al., 2024). Future warming will promote

more surface melt (Trusel et al., 2015), firn air depletion (Kuipers Munneke et al., 2014; Veldhuijsen et al., 2024) and thereby,
possible ice-shelf instability.

Remote sensing is a practical way to monitor surface melt across the vast Antarctic Ice Sheet. Passive-microwave radiometry exploits the strong contrast in brightness temperature between wet and dry snow (Zwally and Gloersen, 1977). It is a powerful technique to observe surface melt year-round, and at high temporal resolution. The penetration depth of the microwave signal varies strongly with frequency — only a few centimetres at 37 GHz ($\sim$2 cm), and increasing up to $\sim$1.8 m at 1.4 GHz — so that each channel samples a different layer of the snow/firn column (Colliander et al., 2022). Several studies have introduced binary melt-day detection approaches based on simple thresholds or polarization and spectral indices to identify liquid water (Zwally and Fiegles, 1994; Abdalati and Steffen, 1997; Torinesi et al., 2003; Picard and Fily, 2006). Importantly, all these passive microwave techniques measure the presence of liquid water, rather than the actual physical process of surface melt (de Roda Husman et al., 2022). In line with common practice in the remote sensing community (e.g., Torinesi et al. 2003, Trusel et al. 2013, Leduc-Leballeur et al. 2020, Banwell et al. 2023), we will interpret the presence of liquid water as snowmelt occurrence, even though liquid water can be present in the snow without melt occurring at the sub-surface. From this point onward, we will refer to observations of liquid-water presence — whether derived from passive-microwave data or from in situ AWS measurements — collectively as "surface melt days."

These approaches provide valuable insights into the spatial and temporal distribution of melt days but do not directly yield water-equivalent melt volumes. A smaller but growing body of work has tackled the challenge of quantifying melt volumes from satellite data. Trusel et al. (2013) empirically calibrated active-microwave QuikSCAT Ku-band backscatter against AWS energy-balance estimates to produce continent-wide melt-volume maps at $\sim$4.5 km resolution. Unfortunately, the QuikSCAT mission ended in 2009. After that, optical satellite imagery has been used to estimate surface melt volumes (Banwell et al., 2021). Other efforts to quantify surface melt volume since then rely on model-based training data. For example, Zheng et al. (2022) used a neural network trained on modelled surface melt to estimate daily melt over Greenland from passive-microwave data at 3.125 km resolution. Banwell et al. (2023) combined passive-microwave and ASCAT scatterometer melt-day counts with the SNOWPACK firn model to derive meltwater volumes on Antarctic ice shelves, for the period 1980–2021, on a 25-km grid.

In this paper, we present the first method to estimate Antarctic melt-volume from passive microwave data that is calibrated solely against melt rate derived from in situ AWS surface energy balance (SEB) observations, and we use this method to produce a continent-wide annual surface-melt rate dataset at 6.25 km resolution for the period 2011–2021. We employ 19 GHz brightness temperatures from SSMIS on DMSP-F17, chosen for high sensitivity to small amounts of liquid water in the snowpack (de Roda Husman et al., 2022) and continuity with earlier SSM/I instruments which potentially enable long-term monitoring. Melt-day occurrence and the melt-day-to-volume relationship are both calibrated directly to melt volumes from seven AWS sites in Antarctica (Van Tiggelen et al., 2025; Jakobs et al., 2020). By using in situ observations for calibrating the satellite signal to melt volume, we indirectly incorporate critical physical feedbacks in the interaction between the snowpack and the atmosphere, such as temperature–albedo interactions (Jakobs et al., 2020), or refreezing dynamics. This multi-tiered

approach — combining high-resolution SSMIS retrievals, AWS-SEB calibration, and model intercomparison — delivers a reproducible, quantitative baseline for Antarctic surface melt rate and identifies pathways for future methodological refinements.

## 2    Materials

This study relies on two main sources of data: satellite-derived brightness temperature from the SSMIS sensor and in-situ observations from AWS. These datasets are used for melt detection, calibration, and validation. The following subsections describe their characteristics and processing. We use the MEaSUREs Antarctic Boundaries Version 2 dataset (Mouginot, 2017) as the Antarctic mask. We define each Antarctic hydrological year as running from 1 June through 31 May of the following calendar year. Accordingly, we analyse ten hydrological years spanning 1 June 2011 to 31 May 2021, corresponding to melt years 2011–12 through 2020–21. Throughout the manuscript, we therefore refer to the temporal coverage as 2011–2021, which reflects the actual range of hydrological years included. This temporal window reflects the overlap between AWS data availability and stable SSMIS observations, which together constrain the coverage of the calibrated dataset.

### 2.1    SSMIS brightness temperature

This study uses brightness temperature from SSMIS on the DMSP-F17 satellite over the hydrological years 2011–2021. DMSP-F17 was selected for its sun-synchronous, dawn–dusk orbit stability, which provides two consistent Antarctic overpasses per day at approximately 06:00 (hereafter "M", morning observation) and 18:00 (hereafter "E", evening observation) local time[1]. All brightness temperatures were obtained from the National Snow and Ice Data Center (NSIDC)[2] and preprocessed in Google Earth Engine (Gorelick et al., 2017). Our analysis concentrates on the H polarized 19 GHz channel, offering a 6.25 km × 6.25 km enhanced footprint — the finest available at this frequency (Brodzik et al., 2024). This channel is widely used for melt detection because 19 GHz is sensitive to small amounts of liquid water while still penetrating into dry firn, yielding low brightness temperatures under dry-snow conditions and a marked increase when liquid water is present (Zwally and Gloersen, 1977; de Roda Husman et al., 2022). We also evaluated the 37 and 91 GHz channels at their enhanced resolutions (3.125 km), but these higher-frequency channels, characterised by much shallower penetration depths (Colliander et al., 2022) did not provide a consistent improvement in our methodology for melt detection.

### 2.2    Automatic weather stations observations

AWS observations are the foundation for the method in this paper. For the melt volume to be calculated, only AWSs that measure sufficient variables to close the surface energy balance qualify. This grossly reduces the number of available AWS locations, since the full radiation budget is only measured at a handful of stations in Antarctica. A major provider of data for this study is the Institute for Marine and Atmospheric Research Utrecht (IMAU) AWS dataset, which is described in Van Tiggelen et al. (2025). Only the IMAU AWSs with at least one entire hydrological year of data within the June 2011 – May 2021 window

---

[1]https://www.remss.com/missions/ssmi/
[2]https://nsidc.org/data/nsidc-0630/versions/2, last accessed 6 June 2025

were used for calibration and evaluation. These comprise AWS11 (Halvfarryggen Ice Rise), AWS14 (northern Larsen C ice shelf), AWS15 (central Larsen C ice shelf), AWS16 (Princess Elisabeth station), AWS17 (Scar Inlet as a remnant of Larsen B ice shelf) and AWS18 (Cabinet Inlet on western Larsen C ice shelf)(Fig. S1). All six sites record the standard meteorological variables and the four components of net surface radiation, with measurements corrected for common errors as detailed in Van Tiggelen et al. (2025). Melt volumes are subsequently computed at each station using the SEB model of Jakobs et al. (2020). In this framework, turbulent fluxes are calculated using similarity theory, surface temperature is determined via iterative closure of the SEB, and excess energy at 0°C is converted into meltwater. Meltwater percolates through the firn using a bucket scheme until refreezing occurs. Shortwave radiation penetration into the subsurface layers of the snowpack is neglected. Of the six IMAU AWS stations meeting our requirements, four (AWS14, AWS15, AWS17 and AWS18) are situated on or immediately adjacent to Larsen C ice shelf, whereas the remaining two (AWS11 and AWS16) provide a few years of measurements in locations with lower melt. We augment the dataset with a decade (2011–2021) of measurements from the German Neumayer station. Although Neumayer also exhibits generally low melt rates, its continuous and long-term record substantially strengthens the calibration dataset and introduces a well-sampled coastal East Antarctic climate distinct from the high-melt conditions of Larsen C. At Neumayer, we use the surface radiation observations from the Baseline Surface Radiation Network (BSRN) station (Schmithüsen, 2021), meteorological observations (Schmithüsen, 2023a), and surface height observations (Schmithüsen, 2023b).

For additional analysis, we also use observations of near-surface air temperature scaled to a nominal height of 2 m above the surface.

Modelling surface melt in an SEB model carries uncertainties because of model settings, model assumptions, and errors in the input. This uncertainty is estimated using a number of sensitivity tests. First, the uncertainty from the IMAU AWS forcing is estimated by separately including or removing one of four measurement corrections: the window heating of the pyrgeometer, the shortwave heating of the passively ventilated temperature sensor, the correction for relative humidity for ice and sensor sensitivity at very low temperatures, and the correction for tilt and bias of the pyranometer, which are all described by Van Tiggelen et al. (2025). To constrain the uncertainty associated with the SEB model, five different model settings were individually adjusted: i) the sensor height fixed to 2m above the surface instead of varying in time, ii) the roughness length for momentum increased from 0.1 mm to 1 mm, iii) the surface longwave emissivity decreased from 1 to 0.97, iv) the snow thermal conductivity parameterised after (Anderson, 1976) instead of (Calonne et al., 2019), and v) allowing the snow height to freely evolve in the model instead of being prescribed by surface height observations. These choices result in one reference and nine perturbed time series of SEB components and surface melt per IMAU station, where each perturbed timeseries results from just one omitted measurement correction or one different model parameter at the time. This sensitivity analysis was only conducted for the AWS where the uncertainty of the observations and of the SEB model are both expected to impact the melt volume computations. These are the AWS that are left unattended for a year or more and located in areas with substantial melt, namely AWS14, AWS15, AWS17, and AWS18.

## 3 Methods

We derive the occurrence of a surface melt day and annual melt totals over Antarctica in two steps. First, we calibrate SSMIS brightness temperature against in-situ surface melt observations at AWS locations to identify robust thresholds that discriminate surface melt from non-melt days (Sec. 3.2). Second, we translate SSMIS-derived melt-day counts to a water-equivalent surface melt volume using an empirical relation derived from the AWS observations (Sec. 3.3).

### 3.1 Melt homogeneity

To assess whether the 6.25 km $\times$ 6.25 km resolution of an SSMIS pixel is sufficient to represent melt conditions at each AWS site, we compared it against the higher-resolution U-Melt binary melt product (de Roda Husman et al., 2024), available at 500 m spatial resolution. For each station, a 13 $\times$ 13 grid of U-Melt pixels was centered over the AWS location, and the melt/no-melt state of all surrounding pixels was compared to that of the central pixel for all days, including both melt and non-melt days.

Two metrics were computed: (i) *homogeneity rate*, defined as the fraction of surrounding pixels with melt flags matching the central pixel, which exceeded 98% at all stations; and (ii) *local variability*, defined as the standard deviation of binary melt values within the window, which remained below 0.02.

These results indicate that, around each AWS, the nature of melt conditions is highly homogeneous at a scale similar to that of the SSMIS pixel footprint. Therefore, we conclude that the 6.25 km $\times$ 6.25 km resolution of the SSMIS pixel is sufficient to represent local melt conditions and is appropriate for calibration purposes.

### 3.2 SSMIS Melt Detection: Calibration and Flagging

To translate SSMIS brightness temperatures ($T_b$) into surface melt-day detections, we assembled a suite of candidate indicators drawn from established microwave-based methods and calibrated each against in situ AWS melt observations ($\geq 0.5$ mm w.e. day$^{-1}$). This threshold was applied to avoid labelling negligible melt amounts, often within the numerical noise of SEB-derived melt estimates, as true melt events, since very small daily values may reflect model uncertainty rather than physically meaningful surface melt.

All metrics were computed at 19 GHz, 37 GHz, and 91 GHz, using both horizontal (H) and vertical (V) polarizations. The indicators were grouped as follows (see Table S1 for a detailed description of all candidate variables):

1. *Pure Brightness-Temperature:* We tested absolute $T_b$ at each frequency and polarization, for both morning and evening observations.

2. *Winter-Anomaly:* Difference between the $T_b$ and its winter mean (Zwally and Gloersen, 1977).

3. *Diurnal and Day-to-Day Change:* i) Diurnal amplitude: difference in $T_b$ between evening and morning overpasses (Ramage and Isacks, 2002). ii) Day-to-day change: difference in $T_b$ between consecutive days at the same overpass time, following approaches similar to those used in short-term $T_b$ variability melt detection (Wang et al., 2016).

4. *Normalized Polarimetric Ratio (NPR):* Contrast between V and H polarizations at the same frequency and overpass (Mousavi et al., 2021).

5. *Normalized Seasonal Anomalies:* Indicators that account for seasonal variability by comparing $T_b$ to its winter anomaly plus a multiple of the winter or annual standard deviation (Torinesi et al., 2003).

Each candidate indicator's day-by-day values were compared against AWS-derived melt versus non-melt classifications. Receiver Operating Characteristic (ROC) analysis was performed on all candidates (Fig. S2), and thresholds were chosen to achieve an optimal trade-off between true positive rate (TPR) and false positive rate (FPR). The two best-performing metrics were the 19 GHz H polarization evening brightness temperature, $T_{b,19\mathrm{H}}^{(E)}$ (TPR $\approx$ 62%, FPR $\approx$ 2%), and the winter anomaly (TPR $\approx$ 67%, FPR $\approx$ 3%). All other candidates yielded TPR below 50%.

### 3.2.1 Multivariate Optimization

Since no single indicator achieved both high TPR and true negative rate (TNR; i.e., 1 - FPR) we selected triplets from the analyzed metrics and applied logical rules i) *and* (all three thresholds must be exceeded for a melt day to be detected); ii) *or* (at least one threshold must be exceeded); iii) *majority* (at least two thresholds must be exceeded) to their thresholds. In 1,000 Monte Carlo trials (randomly sampling 30% of melt and 30% of non-melt days), the *majority* rule achieved the highest overall accuracy and the resulting thresholds exhibited near-Gaussian distributions (Fig. S3). The optimal threshold combination under the majority rule is:

$$\{T_{b,19\mathrm{H}}^{(E)} > 219.2\,\mathrm{K},\ A_w > 26.3\,\mathrm{K}\ \Delta T_d > 19.7\,\mathrm{K},\},$$

where

$$A_w = T_{b,19\mathrm{H}}^{(E)} - \mu_{\mathrm{winter}},$$

is the winter anomaly, with $\mu_{\mathrm{winter}}$ representing the mean 19 GHz H-polarization brightness temperature over 1 June–31 August, and

$$\Delta T_d = T_{b,19\mathrm{H}}^{(E)} - T_{b,19\mathrm{H}}^{(M)},$$

denotes the diurnal amplitude (difference between evening (E) and morning (M) overpasses).

This triplet yields 95.3% accuracy (TPR = 77.8%, TNR = 97.2%), thus balancing false positives and false negatives. Importantly, because negative samples greatly outnumber positive ones in our dataset, a 3% drop in TNR (i.e., more false positives) produces an absolute error count roughly equivalent to that resulting from a 22% drop in TPR (i.e., more false negatives). This analysis is conducted on an annual basis, and the balanced trade-off between false positives and false negatives is achieved at this temporal scale; applying the same thresholds over shorter periods may lead to a disproportionate increase in one error

type. At annual temporal resolution, a 3% decrease in TNR produces an absolute error count comparable to that from a 22% decrease in TPR, demonstrating a balanced trade-off between the two error types at this scale.

### 3.2.2 Melt-Day Flagging and Annual Summation

These three criteria were applied to each set of twice daily SSMIS overpasses for each pixel. A pixel is flagged as "melt" on day $d$ if at least two thresholds are met. SMISS-derived annual melt-day counts are obtained by summing these daily flags per pixel over an Antarctic year (1 Jun to 31 May). A linear regression between AWS-derived and SSMIS-derived annual melt-day counts yielded a coefficient of determination of $R^2 = 0.91$ (Fig. 1a).

### 3.3 Melt estimation

The second major step in the melt volume estimate is to relate the annual number of melt days ($m$) to total annual melt ($M$). To that end, we fitted the AWS-derived decadal record (2011-2021) to an exponential model:

$$M = a\big(e^{bm} - 1\big), \tag{1}$$

where parameters $a$ and $b$ were estimated using a least-squares approach to minimize the residuals between the model and the observed melt values (Fig. 2a). The above functional form follows an empirically demonstrated non-linear relationship between melt days and meltwater production (Banwell et al., 2023; Trusel et al., 2013). This non-linear behaviour likely reflects melt-albedo feedbacks, and the longer time required for refreezing of larger melt volumes, such that warmer summers produce disproportionately more runoff (Banwell et al., 2023). By fitting $a$ and $b$ from equation 1 to AWS stations, which compute melt via a full SEB model, our approach embeds these physical feedbacks into the SSMIS-derived, AWS-calibrated framework.

When the exponential model is applied to the satellite-derived melt-day count, a pixel-level estimate of total annual melt is obtained. A Monte Carlo-based confidence interval for the $m$–$M$ relationship is derived by propagating measurement and model uncertainties (detailed in Sec. 2.2, see Fig. 2a): for each AWS-year combination, ten $m$–$M$ pairs corresponding to distinct setups are available, and in each of 1000 Monte Carlo iterations one setup is randomly selected for each AWS-year, yielding $n$ data points. The exponential model 1 is then fitted to each sample, producing 1000 realizations of $M(m)$ which are evaluated over $m \in [0, 100]$ to characterize the variability of melt estimates. The light pink band in Fig. 2a represents the $3\sigma$ confidence envelope, the blue line denotes the median-fit relationship, and the red line corresponds to the fit obtained using the reference setup alone.

Fig. S4a shows the site-specific exponential fits at each of the four AWS locations (AWS14, AWS15, AWS17, AWS18) where the sensitivity analysis was conducted (see Sec. 2.2); Fig. S4b presents the combined fit across the selected four AWS stations, illustrating how the ten SEB-model permutations produce a modest spread in the resulting $m$–$M$ curves.

For an independent assessment of the $m$–$M$ relation, it was also derived for fully independent, model-only, RACMO2.4 melt-day and melt-volume output for 2011–2021, both across the entire Antarctic domain (Fig. 2b) and separately at four selected AWS locations (see Fig. S4c). In both cases, the resulting exponential parameters and curve shape closely matched

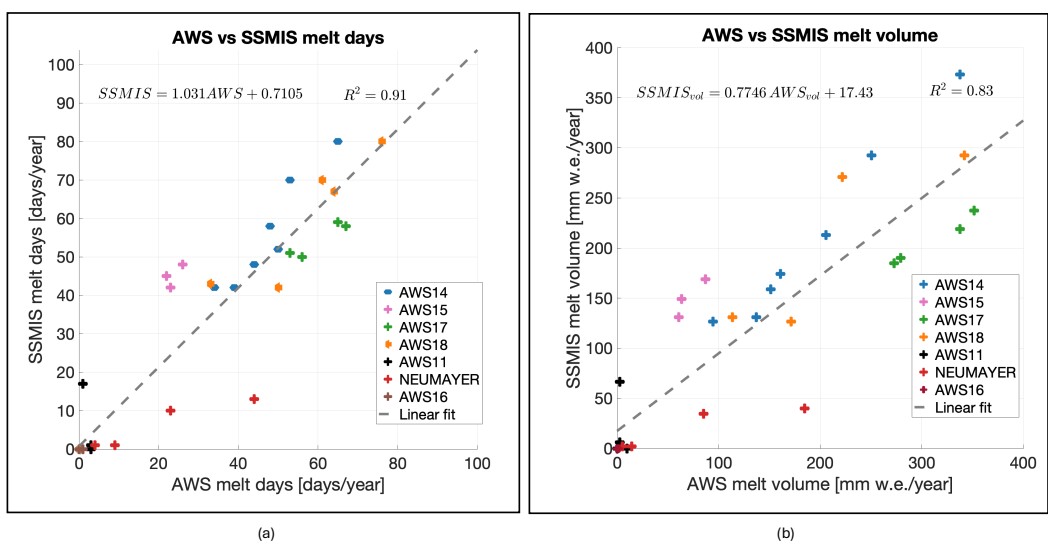

**Figure 1.** Evaluation of all available coincident SSMIS- and AWS-based (a) melt days and (b) melt fluxes across the seven AWS locations.

those derived from the AWS-SEB calibration, demonstrating the robustness and spatial generality of the $m$–$M$ relationship. This also demonstrates that the collection of AWS observations used for this study sufficiently captures the variability in surface melt conditions across the Antarctic Ice Sheet as represented by a physically-based model. The agreement in functional shape, despite the melt days underestimation by SSMIS, supports the application of the AWS-derived fit to satellite-derived melt-day counts across the full Antarctic dataset.

## 4 Results

### 4.1 SSMIS-AWS comparison

Applying the fit described in Section 3.3, we produced estimates of total annual melt across Antarctica (Fig. 3a). We assessed our results by comparing annual SSMIS-derived melt days and melt fluxes with coincident AWS-based observations, yielding a strong linear correlation ($R^2 = 0.91$ and $R^2 = 0.83$, respectively; Fig. 1b). However, given the limited number of in situ AWS sites — which were also employed during calibration — this evaluation is inherently circular. Dividing the AWS record into independent calibration and validation subsets was considered not feasible due to the small sample size and the constrained spatial variability of the available stations.

### 4.2 Comparison of SSMIS with QuikSCAT and RACMO2.4

We compare our ten-year decadal-mean melt-flux estimates from SSMIS with two independent products:

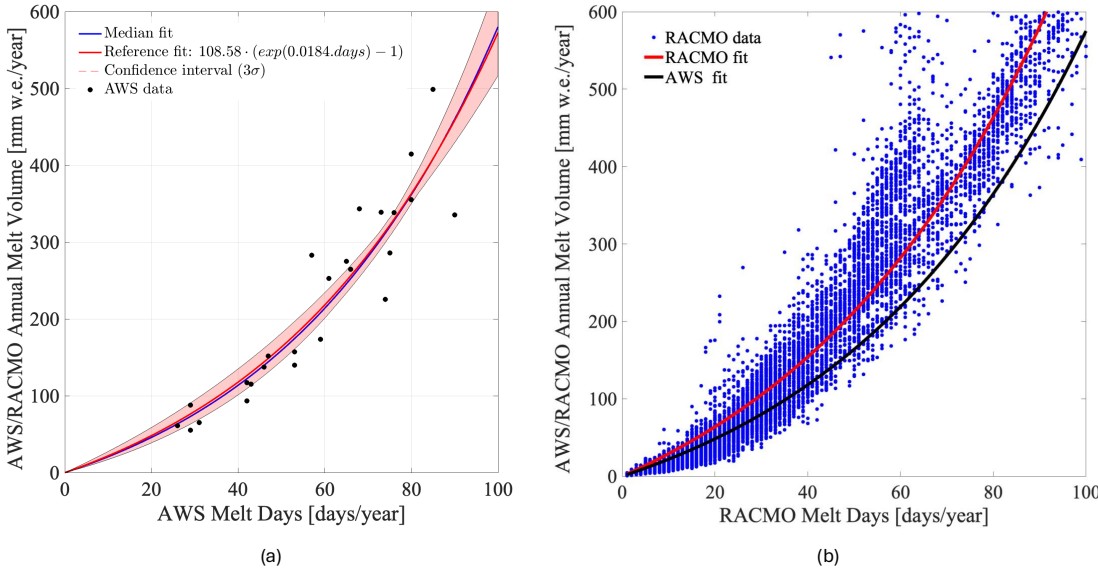

**Figure 2.** Exponential melt-day to melt-volume relationship. **(a)** Scatterplot of annual meltwater volume ($M$, from AWS-SEB) versus melt days ($m$) at six IMAU-AWS stations for 2011–2021, with the best-fit exponential curve shown in red ($R^2 = 0.91$). The median fit from 1,000 Monte Carlo realizations is shown in blue, and the shaded pink band indicates the $\pm 3\sigma$ confidence interval. **(b)** Comparison of the AWS-derived $m$–$M$ curve (black) against RACMO2.4 ($R^2 = 0.91$): the red line is the RACMO2.4 fit, while blue dots represent RACMO2.4 pixel-level data for all of Antarctica over 2011–2021.

- **QuikSCAT (1999–2009)**: decadal-mean annual melt flux derived from Ku-band backscatter at 4.45 km resolution (Trusel et al., 2013), see Fig. 3b.

- **RACMO2.4p1 (hereafter, RACMO2.4) (2011–2021)**: decadal-mean annual melt flux simulated at 11 km resolution (van Dalum et al., 2025), see Fig. 3c.

Across the Antarctic Peninsula, all three datasets show consistently high decadal-mean surface melt rates. On Larsen C Ice Shelf, SSMIS, QuikSCAT, and RACMO2.4 all exceed 350 mm w.e. yr$^{-1}$. SSMIS and QuikSCAT place their highest decadal-mean melt values along the western inlets (e.g. Mill Inlet), whereas RACMO2.4 shifts its maximum eastward toward Scar Inlet, a spatial offset also noted in earlier satellite-based analyses (Trusel et al., 2013). Farther south, on Wilkins and George VI ice shelves, decadal-mean melt rates exceed 200–250 mm w.e. yr$^{-1}$ in all datasets.

Along coastal West Antarctica, including the Amundsen and Ross Sea sectors, decadal-mean melt rates remain low, around 20–30 mm w.e. yr$^{-1}$ in all products. These areas represent some of the lowest-melt regions outside the high-elevation interior.

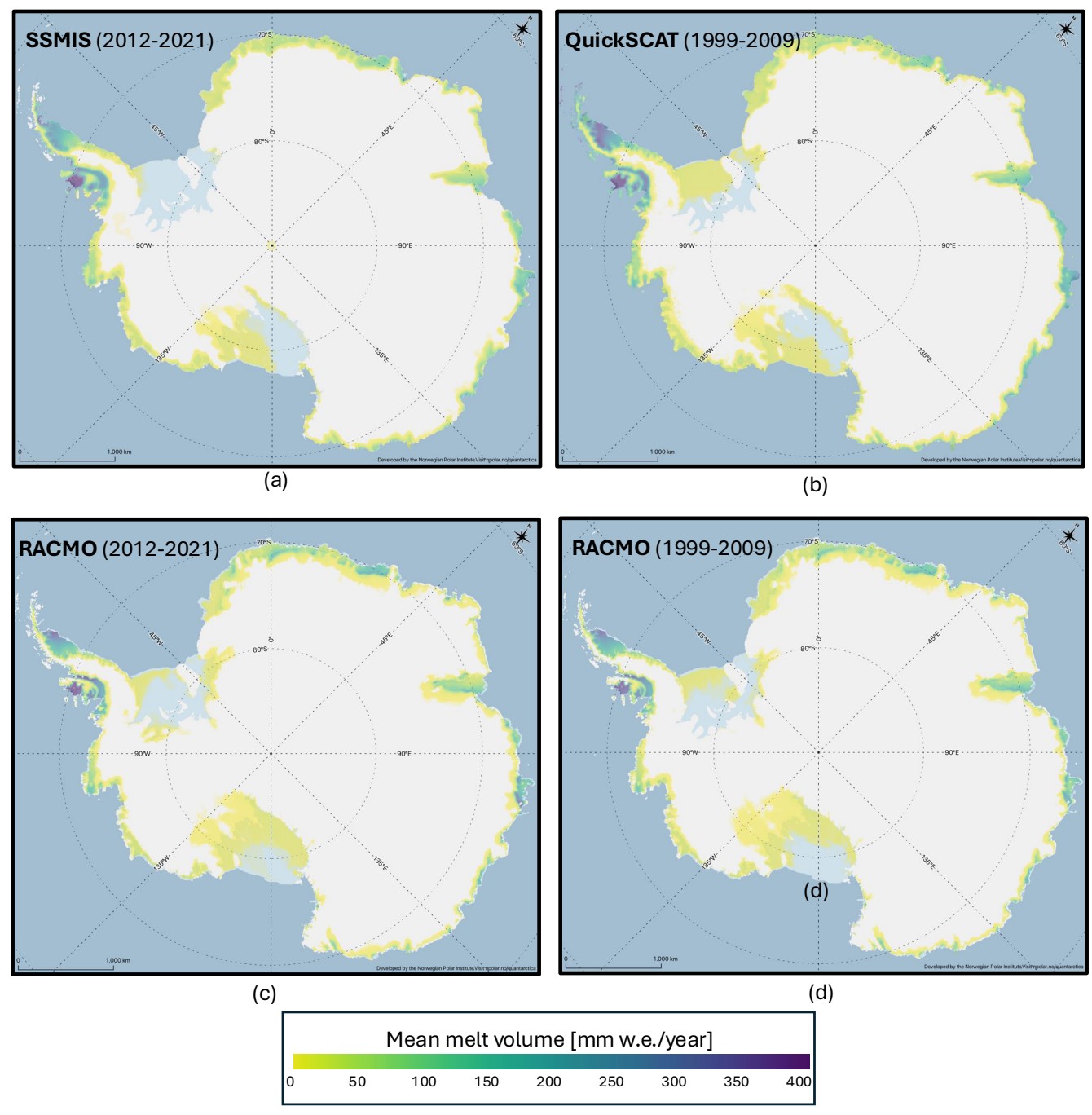

**Figure 3.** Comparison of decadal mean meltwater volume across Antarctica. (a) SSMIS-derived annual melt flux averaged over 2011–2021. (b) QuikSCAT-derived melt flux over 1999–2009 from backscatter observations (Trusel et al., 2013). (c) RACMO2.4 model output averaged over 2011–2021 (van Dalum et al., 2025). (d) RACMO2.4 model output averaged over 1999–2009 (van Dalum et al., 2025).

Over the Ross Ice Shelf, SSMIS and QuikSCAT show the strongest decadal-mean melt along the western flank, whereas RACMO2.4 simulates higher melt along the eastern margin.

In East Antarctica, the three products again show broadly consistent patterns. Decadal-mean surface melt rates of around 200 mm w.e. yr$^{-1}$ occur on Roi Baudouin and the inner Fimbul ice shelves, while northeast Amery Ice Shelf shows rates near 150 mm w.e. yr$^{-1}$ across all products.

To allow a direct comparison with QuikSCAT, we also extracted RACMO2.4 outputs for 1999–2009 (Fig. 3d). Over this shared period, the two datasets show comparable decadal-mean melt magnitudes (within ∼10%) and similar spatial patterns, including RACMO2.4's modest eastward displacement of melt maxima and QuikSCAT's tendency to underestimate melt in low-intensity coastal regions. These agreements and discrepancies are consistent with those observed between SSMIS and RACMO2.4 for 2011–2021.

Interannual melt volumes from SSMIS and RACMO2.4 over 2011–2021 exhibit similar temporal patterns. The mean annual Antarctic melt volume is approximately 85 Gt yr$^{-1}$ for SSMIS and 100 Gt yr$^{-1}$ for RACMO2.4 (Fig. S5). The corresponding annual mean melt-flux maps (Fig. S6) further demonstrate the close spatial and temporal agreement between the two datasets, and additionally provide a regional view of the Antarctic Peninsula, where most Antarctic surface melt occurs.

## 5   Discussion

A closer look at misclassified surface melt days reveals two primary sources of false positive detections. About 71 % of false positives (defined here as days classified as melt by SSMIS while AWS-SEB reports zero melt) occur when RACMO2.4 simulates liquid water content (LWC) in the firn (See Fig. S7a). Nearly 90 % of false positives coincide with AWS near-surface air temperatures ($T_{2m}$) above –5 °C (See Fig. S7b). Taken together, these patterns indicate that the classifier is responding to liquid water within the near-surface firn, even when surface melt is not diagnosed by AWS-SEB. This behaviour is consistent with the known penetration depth of 19 GHz microwave radiation, which is sensitive to both surface and shallow subsurface wetting (de Roda Husman et al., 2022). In this sense, SSMIS detects a broader physical melt–wetting signal that includes processes not directly measurable by AWS but captured by RACMO2.4's subsurface hydrology. For this reason, we also explored the potential of additional microwave indicators, such as the 37 GHz channel and various polarization or spectral ratios, to reduce false positives by improving sensitivity to surface wetting. While these metrics offer theoretical advantages due to their shallower penetration and enhanced surface melt response (Colliander et al., 2022), our cross-validation results show no consistent performance improvement across the AWS network. This outcome supports our choice of the 19 GHz H-polarization channel as the most robust and spatially consistent indicator under current sensor constraints. A closer examination of Fig. 1 shows that Neumayer station exhibits larger residuals than the other sites. This discrepancy likely reflects Neumayer's local climate, where subfreezing daytime temperatures drive nearly instantaneous firn refreezing (van den Broeke et al., 2010). Consequently, less liquid water remains at the surface during SSMIS overpasses, diminishing the brightness-temperature signal compared to other AWS locations — such as Larsen C — where subsurface water retention prolongs wet-snow signatures. Although the SSMIS dataset covers all of Antarctica, the calibration relies on a geographically limited set of AWS sites, with

four stations located on Larsen C and only three additional sites elsewhere. This raises the question of whether the melt–day to melt–volume parametrisation is transferable across the full Antarctic melt zone. However, two lines of evidence suggest that the calibration is broadly representative: (i) the sensitivity analysis combining all AWS years produces a stable, well-constrained $m$–$M$ relationship, and (ii) RACMO2.4 exhibits a nearly identical functional relationship across the entire ice sheet (Fig. S4c). These comparisons indicate that, despite the sparse calibration network, the underlying exponential relation is sufficiently general to apply across contrasting climatic regions, though local deviations cannot be fully excluded.

From a spatial perspective, our melt product reveals interesting regional features. For instance, on the Larsen C ice shelf, a distinct east-west gradient is visible, likely driven by föhn winds over the Antarctic Peninsula mountain range (Luckman et al., 2014) and supported by melt patterns in QuikSCAT (Trusel et al., 2013), and firn air content observations across the ice shelf (Holland et al., 2011). The SSMIS-based method shows less surface melt relative to QuikSCAT — but the first was collected a decade after the second. Thus, its difference may be attributed to the documented cooling trend over the Peninsula in the first decade after 2000 (Turner et al., 2016), which has been linked to decadal-scale natural climate variability. Taken together, our findings suggest that the proposed SSMIS-based detection scheme reasonably captures the spatial and temporal patterns of surface melt across Antarctica. Its general consistency with known climate trends indicates that the classifier is likely robust to both environmental variability and regional melt characteristics. However, the sensitivity to shallow wetting layers — while offering valuable insight into subsurface processes — also introduces uncertainty when interpreting daily melt flags. Refining this ambiguity represents a necessary direction for improving the distinction between surface melt and the presence of sub-surface liquid water in future satellite-based algorithms.

## 6 Conclusions

We introduce a novel 6.25 km gridded dataset of Antarctic surface melt rates for 2011–2021, derived exclusively from SSMIS 19 GHz passive-microwave observations and calibrated against seven AWS energy-balance melt records. Our majority-rule framework —combining absolute evening $T_b$, diurnal amplitude, and winter-season anomaly thresholds — yields daily melt flags that, when transformed through an exponential melt-day to melt-volume model, reproduce in-situ melt volumes with fidelity. Comparative analyses with QuikSCAT and RACMO2.4 confirm that our product accurately maps melt hotspots, while misclassification analysis clarify the conditions under which passive-microwave retrievals are least reliable.

By providing a spatially comprehensive SSMIS-derived, AWS-calibrated record of Antarctic surface melt, this dataset fills a critical gap between sparse in-situ measurements and model outputs. It offers a transparent, reproducible baseline for evaluating regional climate models, constraining firn-hydrology schemes, and informing assessments of ice-shelf vulnerability to meltwater-induced weakening. The complete Antarctic-wide, decadal melt record is publicly available for use in cryospheric process studies.

*Code and data availability.* The annual Antarctic surface melt–water equivalent maps derived from SSMIS 19 GHz brightness temperatures, covering the period 2011–12 to 2020–21, are publicly available at https://doi.org/10.5281/zenodo.16738423 (Di Biase, 2025). The dataset includes GeoTIFF files providing annual number of melt days and cumulative annual melt volume per pixel (in mm water equivalent) with corresponding lower/upper bound estimates based on the confidence intervals represented in Fig.2(a) to convey the uncertainty range.
The AWS data used as forcing for the SEB model is available at https://doi.pangaea.de/10.1594/PANGAEA.974080 (Van Tiggelen et al., 2024). The SEB model used to compute surface melt is available at https://doi.org/10.5281/zenodo.15082295 (Van Tiggelen et al., 2025).

*Author contributions.* VDB developed the methodology, carried out validation, performed the formal analysis and investigation, curated the SSMIS data, and wrote the original draft. MvT contributed to the study conceptualization, managed AWS data curation, review and editing of the manuscript. PKM, BW, and MvdB contributed significantly to the overarching conceptual framework, supervised the research, the review and editing of the final manuscript.

*Competing interests.* B.W. and M.vdB. are members of the editorial board of journal The Cryosphere.

*Acknowledgements.* The authors would like to thank Christiaan van Dalum for providing the RACMO2.4 dataset, which was instrumental in the comparative analyses presented in this study. The authors would also like to thank the members of the Institut des Géosciences de l'Environnement (IGE, Grenoble Alpes, France) for the valuable suggestions and fruitful exchanges throughout the development of this work. M.vdB. acknowledges funding from EMBRACER (Summit Grant SUMMIT 1.034), financed by the Netherlands Organisation for Scientific Research (NWO).

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
