# Peer review of "Estimating Antarctic surface melt rates using passive microwave data calibrated with weather station observations."

_EGUsphere, 2025_

## Referee Comment (RC1)

Review of Brief Communication: Estimating Antarctic surface melt rates using passive microwave data calibrated with weather station observations

This manuscript presents a novel dataset of continent-wide Antarctic annual surface melt rates spanning from June 2011 to May 2021. The authors derive an empirical relationship between annual melt days and annual melt volume from a surface energy balance model driven by seven ground based automatic weather stations located on the Antarctic Peninsula and across Dronning Maud Land, East Antarctica. Annual surface melt rates are then derived across the whole of Antarctica using the empirical equation and imagery obtained from the passive microwave sensor SSMIS, which offers year-round detection of melt in both the surface and near-surface layers of the snowpack. Similar datasets of empirically derived surface melt rates from microwave imagery do exist, however this work presents the first dataset of melt rates derived from passive microwave imagery and builds on previous work by offering an extended temporal coverage.

The manuscript is generally well-written with a logical structure that is easy to follow and clear figures that supplement the manuscript's methodology and results nicely. In some cases, small changes would improve the clarity of the manuscript even further, which I have outlined in my comments below. I have some questions but no major concerns regarding the methodological approach, which follows well-established methods and is proven to be robust within the analysis.

**Major comments**

1. Currently, the introduction provides a good initial overview of microwave-based methods for identifying and quantifying snowmelt, and therefore the knowledge gap which this study contributes to. However, I think there is key context missing from the introduction, both in terms of driving home the novelty and wider importance of this study, as well as key themes utilised within the methodology. Below are my suggestions:

   *Wider context:*
   - L13: Add more recent citations in addition to the papers already cited.
   - I think the introduction should outline the relationship between the firn air content/depletion of the snowpack (see Dunmire et al. (2024) - https://www.nature.com/articles/s43247-024-01255-4), snowmelt, and surface melt ponding.
   - I appreciate L13 mentions the role of surface melt in ice-shelf collapse, but I think the role of surface melt in hydrofracture should be explicitly made.

   *Study methodology:*
   - Given that the empirical derivation of surface melt rates from AWS surface energy balance modelling is a key component of this study's methodology, I think it is important to introduce the concept of the surface energy budget and its role in driving snowmelt. This would also make the significance of the critical physical feedback mentioned on L42 clearer.

- I think the benefits of microwave-derived melt detection should be mentioned, namely year-round, higher-temporal resolution detection of surface and subsurface melt.

*Other comments:*
- I noticed L20-24 were very similar to a section of de Roda Husman et al.'s (2022) paper (see below). The authors cite de Roda Husman's paper, but I would encourage the authors to rephrase what they have written to avoid it being too similar to previous work. I'd encourage the authors to check other parts of their manuscript in case this issue is more widespread.

  Extract from de Roda Husman et al. (2022), page 2463:
  "*These parameters therefore are indicators for the presence of liquid water, rather than for the actual physical process of surface melt, which is in fact an energy (conversion) process, except, perhaps, in the case of thermal-infrared-derived surface temperature that defines the occurrence of surface melt at the melting point. Yet, the term "surface melt" is widely used in the remote sensing community (e.g., [10], [16], [18], [20]), and we will adopt it here, although we do acknowledge that sensors measure the presence of liquid water.*"

  Lines 20-24 from this manuscript:
  "*These approaches are indicators for the presence of liquid water, rather than for the actual physical process of surface melt, which is in fact an energy-conversion process (de Roda Husman et al., 2022). Yet, the term "surface melt" is widely used in the remote-sensing community (e.g., Torinesi et al. 2003, Trusel et al. 2013, Leduc-Leballeur et al. 2020, Banwell et al. 2023), and we will adopt it here, although we acknowledge that passive-microwave sensors detect liquid water in the snowpack, independently of whether active melting is occurring at the surface.*"

  - L27 - L31: Surface melt volumes have been derived from optical satellite imagery in numerous studies – for example, see section 3.3 'Landsat 8 and Sentinel-2 derived meltwater areas and volumes' in Banwell et al. (2021) (https://tc.copernicus.org/articles/15/909/2021/) and Bell et al. (2017) (https://www.nature.com/articles/nature22048), both of which are more recent than 2009.

2. I have three main comments regarding Section 3.1 of the manuscript. First, I think the overall purpose of the method application and analysis could be made clearer by a slight adjustment to the terminology used. I originally thought that this section would assess whether the SSMIS pixels containing the AWSs correctly identified the presence of melt for the same days that the AWSs identified melt, as "accurately" implies comparison against a true value. Therefore, for this section I offer the following suggestions:

- Adjust L105 to something along the lines of *"To assess whether the 6.25 km x 6.25 km resolution of an SSMIS pixel is sufficient to represent melt conditions at each AWS site…"*
- So that it is clear which dataset (SSMIS or UMelt) is being used, adjust L107 to *"For each station, an 11 x 11 grid of UMelt pixels was centred over each AWS location..."*
- For greater clarity, adjust L112-114 to *"These results indicate that around each AWS, the nature of melt conditions is highly homogenous at a scale similar to that of the SSMIS pixel footprint. Therefore, the 6.25 km x 6.25 km resolution of the SSMIS pixel is sufficient to represent local melt conditions and thus is appropriate for calibration purposes."*

Second, it is not clear how each UMelt grid was selected, nor how many were analysed through this process. Was this carried out for one grid per AWS or multiple grids per AWS? How was each respective grid chosen? I'm assuming they were only selected for days when melt was observed, but I think it would be good to state this. A short explanation addressing the above questions would be a good addition to this section.

Finally, I am interested to know why an 11 x 11 grid of UMelt pixels was chosen? If my interpretation of your methodology is correct, using a 13 x 13 grid of 500 m x 500 m pixels would produce an overall footprint of 42.25 $km^2$ which: i) is much closer in size to the ~40 $km^2$ footprint of an SSMIS pixel compared to the 30.25 $km^2$ footprint of an 11 x 11 grid, and ii) has a footprint marginally larger than the SSMIS pixel, such that the SSMIS pixel is fully contained within the UMelt grid. Currently, the melt homogeneity evaluated over the 30.25 $km^2$ (11 x 11) UMelt footprint is reflective of only ~75% of the SSMIS pixel, whereas using a 42.25 $km^2$ UMelt footprint (13 x 13 grid) would enable an evaluation of homogeneity rate and local variability across 100% of the SSMIS footprint. If the homogeneity rate and local variability metrics were computed over a 13 x 13 grid, and produced very similar results, then the conclusion of L112-114 could be strengthened even further, e.g., *"These results indicate that around each AWS, the nature of melt conditions is highly homogenous **at a scale greater than that of the SSMIS pixel footprint**. Therefore, the 6.25 km x 6.25 km resolution of the SSMIS pixel is sufficient to represent local melt conditions and thus is appropriate for calibration purposes."*.

As the purpose of this section is to evaluate whether the enhanced resolution of the SSMIS pixel is sufficient, even if these metrics decrease, I think it is important to provide a quantification of melt homogeneity over the entire SSMIS pixel footprint.

3. From my interpretation of the ROC analysis (Figure S1), it appears the two best-performing metrics stated on L132 have been incorrectly identified. Instead of diurnal amplitude, this should be winter anomaly. This error doesn't have subsequent implications for the following stages of analysis, but L132 should be corrected. In addition, four of the 'Area Under the Curve' (AUC) statistics shown on Figure S1 are reported as negative, which is not possible (an area cannot be negative). This likely results from plotting the False Positive Rate (FPR) in descending rather than ascending order, causing the integration for AUC to yield a negative value. However, the magnitude of AUC will be correct, and

therefore the analysis and interpretation of Figure S1 does not need to be repeated, but the relevant plots should be edited to remove the negative sign. In light of the above, I would also suggest reordering the indicators in L141 and in the following lines (L142-147) so that winter anomaly comes before diurnal amplitude.

**Minor comments**

Throughout the manuscript, the RACMO2.4 dataset used within this study is sometimes referred to as 'RACMO', e.g., L8 or 'RACMO2.4p1' e.g., L197. For continuity and clarity, I would suggest either consistently using 'RACMO2.4' throughout, including within figures, or after the first use of 'RACMO2.4' follow with "hereafter referred to as RACMO".

L30: The acronym 'AWS' has already been established earlier in the manuscript and so "automatic weather station (AWS)" does not need to be written again in full here. Check this here and elsewhere in the manuscript.

L33-34: Would be good here to give spatial resolution and temporal coverage of Banwell et al.'s (2023) dataset.

L35-44: The methodology of this study is introduced but there is no mention of the resulting dataset which I think should be included here.

L50: If AWS data availability is the limiting factor on the temporal coverage of this study's dataset, I think this point needs to be made more explicitly. Surely these AWSs have data available to the present day?

L51: The manuscript states that the temporal coverage of the dataset is "2012-2021". However, here, the authors note June 2011 – May 2021 as the respective start and end dates of the ten hydrological year-period for which the dataset is produced. Given that the dataset offers annual surface melt rates for the hydrological year 2011-2012 (as shown in first panel of Figure S5), would reporting the temporal coverage of the dataset as "2011-2021" be more inclusive of the actual temporal range? It would also clearly show from a quick glance that it's a decadal dataset.

L56-58: As the SSMIS sensor offers both a standard and an enhanced resolution dataset at 19 GHz frequency, the manuscript should state that the enhanced resolution dataset is being utilised and should use the relevant citation for this (Brodzik et al. (2024): https://nsidc.org/data/nsidc-0630/versions/2)

L57: In the interest of accessibility for all audiences, the manuscript could benefit from a clearer justification for the focused use of the 19 GHz H channel (assuming this was selected due to the general acknowledgement of this channel as having the lowest brightness temp over dry firn?)

L58: As with the 19 GHz channel, the 37 and 91 GHz frequencies also offer standard (25 km) and enhanced resolutions (3.125 km), though it is not clear which resolution is utilised here? It would also be helpful to provide an estimate of penetration depth for each of the three channels. This could provide further context to the authors' statement that utilising the 37 and 91 GHz frequencies didn't notably enhance melt detection; i.e., that most of the detected melt occurs within the sub/near-surface layers of the snowpack. This edit could be addressed at this stage of the manuscript or earlier in the introduction (L16-17).

L57-59: Horizontal and vertically polarised channels are referred to a lot throughout manuscript and in equations. Introduce the acronyms 'H' and 'V' here and then use consistently throughout.

L66-68: For readers less familiar with AWSs, it would be helpful to give the region in which they are located either in their respective brackets, e.g., "AWS14 (northern Larsen C Ice Shelf, *Antarctic Peninsula*)" or introduce them by region, e.g., "*Four AWSs from the Antarctic Peninsula (AWS14, …) and three from Dronning Maud Land, East Antarctica (AWS11, AWS16, and Neumayer)…".* A useful addition to this manuscript would be a supplementary Figure showing the exact locations of each station. This would give a reader a quick sense of their location and spatial distribution across the Peninsula and Dronning Maud Land.

L73: Adding *"into the subsurface layers of the snowpack"* here would increase clarity.

L76-78: In line with my above comment, give the location of Neumayer station (Ekström Ice Shelf, Dronning Maud Land, East Antarctica) or illustrate its location in a Figure. Here, the authors include Neumayer station with the purpose of broadening the geographic scope for their model calibration to ensure that it performs robustly across climatically distinct regions. Given that four of the AWSs are located on the Peninsula and the other two stations are also located in Dronning Maud Land, East Antarctica, would the inclusion of station data from further afield, such as the West Antarctic Ice Sheet, offer a more comprehensive addition of data to this study? If this is not possible, an explanation of how the inclusion of Neumayer station, in addition to AWS11 and AWS16 (also located in Dronning Maud Land), provides data from a broader geographic scope and climatically distinct region would support these statements.

L89-92: I find this section a little confusing and hard to follow. Here are my suggested edits:

> "*To constrain the uncertainty associated with the  SEB model,  each of the five model settings were individually adjusted  i) sensor  height was fixed  at 2 m above the surface  ii)  momentum roughness length was increased from 0.1 mm (typical for snow) to 1 mm  iii)  surface longwave emissivity was decreased from 1 to 0.97  iv)  an alternative snow thermal conductivity was used (Anderson, 1976), and  v)  instead of nudging snow height with sonic height ranger observations, it was allowed to freely evolve in the model *"

For point iv), I think this remains too vague -- what alternative was used? For the earlier list of measurement corrections (L85-87), I would recommend also using the i), ii), iii) … notation, as above, for better readability.

L95: Why were sensitivity tests not carried out for all stations? Pointing the reader here to section 3.3 where the sensitivity tests are incorporated and discussed would be beneficial.

L102-103 repeats line 105. Remove L102-103.

L117: Is $\geq 0.5$ mm w.e. day$^{-1}$ the threshold for identifying a "surface melt day" at the AWSs? If so, I think this could be explicitly stated and justified i.e., why is it not just any day where melt rate is $> 0$ mm w.e. day$^{-1}$? Is $\pm 0.5$ mm w.e. the resolution of the AWS measurements?

L124: Is there an appropriate reference that could be cited for the Day-to-Day change? Maybe Wang et al. (2025) (https://tc.copernicus.org/articles/10/2589/2016/tc-10-2589-2016.pdf)? Though I appreciate they compare daily $T_b$ values to a previous-3-day average.

L125: As Abdalati and Steffen (1997) use a cross polarised gradient ratio where both frequency and polarization is evaluated, could a study that uses the normalized polarization ratio be more appropriate? e.g., Mousavi et al. (2021) (https://dspace.mit.edu/handle/1721.1/148558)

L129: Over what time period was this comparison made and, in line with my earlier comment, is AWS-derived melt vs non-melt classified using $\geq 0.5$ mm w.e. day$^{-1}$ as an absolute threshold?

L157: As there are two sets of annual melt-day counts (SSMIS-derived and AWS-derived), for greater clarity be clear which set is being referred to – e.g., "_SSMIS-derived annual melt-day counts are obtained by…_". Likewise, when referring to "observations" e.g., L81, make it clear these are AWS observations.

For Figure 2, it would be good to report an $R^2$ value for both Figure 2a and 2b to quantify the fit of the data. I suggest labelling each respective y axis as "_AWS/RACMO Annual Melt Volume [mm w.e./year]_" for clarity.

L203-205: It would be nice to visually see these results either as a closer view inset on Figure 3 or as an additional Figure in supplementary materials.

Here and throughout the results section, be clear about the exact statistics that are being reported, i.e., L204: "_melt rates exceeded 350 mm w.e. yr$^{-1}$… show their highest values on the western inlets..._" – are these annual melt rates? decadal mean? highest mean values? For L210 (and elsewhere)—"_annual surface melt at Roi Baudouin_"—be clear that these are surface melt rates.

L203-214: Consider presenting your results organised by region (Antarctic Peninsula, West Antarctica, East Antarctica). This will make them easier for the reader to digest. I think it would also be interesting to give regional decadal mean melt fluxes for the Peninsula, WAIS, and EAIS.

L212: Is this region of low-intensity melting a continent-wide minimum for Antarctica or just an area of low melt rates?

L224: It is not clear here which products the melt classification is being evaluated for, and therefore what is considered as a false positive?

L226-231: Surely these discussion points could be reframed with more confidence in the argument presented?

> "*These findings **suggest** that our classifier is not only responding to surface melt events, but more generally detects the presence of liquid water near the surface…*"
>
> The ability of SSMIS to penetrate and detect near/subsurface melt in the snowpack is already acknowledged by the authors early on in the manuscript (L23-24).
>
> "*In this sense, SSMIS **appears** sensitive to a broader melt signal spectrum, including processes not directly measurable by AWS but captured by RACMO's subsurface hydrology.*"
>
> Is SSMIS not already acknowledged for having a broader melt signal given the ability for lower frequency channels to penetrate deeper into the snowpack?
>
> It would be interesting to comment on the implications of neglecting shortwave penetration into the snowpack during the AWS SEB modelling (L73).

L241: This recent preprint by Zou et al. might be of interest here – https://doi.org/10.21203/rs.3.rs-7384193/v1

L245: Turner et al. (2016) identifies the cooling trend from 1998. I also think it would be worth mentioning the cooling trend was associated with decadal-scale natural variability.

L258: Here and elsewhere, the surface melt rate dataset is referred to as "satellite-only/ derived exclusively from SSMIS…". As this study relies not only on satellite-derived brightness temperatures but also AWS observations – as stated by the authors on L46 – I would consider rephrasing from "satellite-only".

Figure S5: Include in figure caption that these are regional maps of the Antarctic Peninsula.

**Technical corrections**

L2: "Antarctic I̲ce S̲heet" – correct this here and elsewhere in the manuscript

L37: "From SSMIS on t̲h̲e̲ DMSP-F17 s̲a̲t̲e̲l̲l̲i̲t̲e̲"

L62: "AWSs̲"

L65: "IMAU AWSs̲ "

L156: "set of  t̲w̲i̲c̲e̲ daily..."

L170: "a pixel-level estimate of t̲o̲t̲a̲l̲ annual melt"

L174: "exponential model  (̲1̲)̲ is then fitted"

L224: "A closer look at misclassified s̲u̲r̲f̲a̲c̲e̲ ̲m̲e̲l̲t̲ days..."

L225: "occur when RACMO2.4p1 simulates  liquid water content…"

L244: "a decade  a̲f̲t̲e̲r̲ the second"

L254: "diurnal amplitude, and winter-season anomaly t̲h̲r̲e̲s̲h̲o̲l̲d̲s̲ – yields…"

---

## Referee Comment (RC2)

**Review of "Estimating Antarctic surface melt rates using passive microwave data calibrated with weather station observations", by Di Biase *et al*. (egusphere-2025-2900).**

**General**

This paper presents a new dataset of annual Antarctic surface melt occurrence and meltwater flux, derived from satellite passive microwave radiometry. The presence of surface melt is detected using a newly-developed algorithm that uses an optimised set of indices derived from the 19 GHz channel of the SSMIS sounder. The algorithm is calibrated using surface energy balance measurements from several automatic weather stations (AWS) in coastal Antarctica. The AWS measurements are also used to tune a parametrisation that estimates annual meltwater flux from the annual sum of melt days. The authors present maps of annual melt frequency and meltwater flux for the period 2012-2021 and compare these with maps derived from an existing melt product and with regional climate model output.

The paper is clearly written and the methodology appears to be sound. I was particularly impressed by the rigorous error analysis that the authors have carried out, which has enabled them to quantify the uncertainties in their product. Their dataset will undoubtedly find application in Antarctic cryospheric and climate science, particularly as a source of independent validation data for regional and global models. While the extent and occurrence of surface melt in Antarctica is currently limited, both spatially and temporally, it is likely to become more widespread in a warming climate. It is thus of vital importance to validate the representation of current surface melt in models that are used to make projections of future melt. In my opinion, the paper is highly suitable for publication in *The Cryosphere* following minor revision as detailed below.

**Main Points**

1. SSMIS data cover the whole of Antarctica, making it possible to generate an Antarctic-wide melt product. However, calibration data are only available for the six AWSs and Neumayer Station. Furthermore, four of the AWSs are located on the Larsen Ice Shelf. The calibration data are thus quite geographically restricted, which begs the question of whether parametrisations derived using these data will be valid across all of the Antarctic melt zone. I think that there are indications that this may be true (e.g., figure S3(c), which, I think, provides model-based evidence that just using these sits for calibration does not introduce major biases) but I'd like to see a bit more discussion of the possible uncertainties resulting from limited calibration data.

2. In section 4.2 you discuss the spatial structure of the melt field in the Antarctic Peninsula but it is very difficult to see at the scale of the pan-Antarctic maps presented in figure 3. Given that most melt occurs in the Peninsula region (and that the majority of your calibration data come from this region) I'd consider presenting separate larger-scale maps covering just the Peninsula region.

3. Not all readers will be familiar with the locations of the AWSs. I think that a location map would be useful.

**Minor Points**

1. Line 14: I'd say "…the only practical way…"

2. Lines 50-51: Make it clear that your "year" label corresponds to the END of the hydrological year.

3. Line 52: For clarity, replace "the period" with "hydrological years".

4. Lines 53-55: Using overpasses at local times 06:00 and 18:00 may miss some melt days if melt is only occurring during the warmest part of the day. You do discuss this (with reference to Neumayer) in lines 237-239.

5. Line 59: replace "it" with "this".

6. Line 71: "similarity theory".

7. Line 234: "supports".

---

## Author Response (AR1)

**Response to Reviewer1**

*We thank the reviewer for the thorough and constructive assessment of the manuscript. Following the editor's advice to change this brief communication into a full paper, we have been able to address the reviewer's comments, including expanding the introduction and adding recent citations. The comments have substantially improved clarity, framing, and methodological transparency. Below, we respond to each point individually. Reviewer comments are reproduced in full, followed by our responses in blue.*

Major comments

1. [Introduction]:

*We thank the reviewer for these constructive and detailed suggestions regarding the Introduction. We have substantially revised this introduction to (i) expand the broader scientific context of Antarctic surface melt, including firn-air depletion, melt ponding, and hydrofracture susceptibility; (ii) introduce the role of the surface energy budget in driving melt and motivating the AWS–SEB calibration used in this study; (iii) highlight the advantages of microwave remote sensing for year-round monitoring of surface and subsurface melt; and (iv) incorporate additional and more recent relevant literature, including optical-satellite estimates of melt volume. In addition, the paragraph on passive-microwave interpretation has been fully rephrased to avoid similarity with de Roda Husman et al. (2022). These revisions collectively strengthen the motivation, framing, and novelty of the study.*

2. I have three main comments regarding Section 3.1 of the manuscript. First, I think the overall purpose of the method application and analysis could be made clearer by a slight adjustment to the terminology used. I originally thought that this section would assess whether the SSMIS pixels containing the AWSs correctly identified the presence of melt for the same days that the AWSs identified melt, as "accurately" implies comparison against a true value. Therefore, for this section I offer the following suggestions:

 • Adjust L105 to something along the lines of "To assess whether the 6.25 km x 6.25 km resolution of an SSMIS pixel is sufficient to represent melt conditions at each AWS site..."
 • So that it is clear which dataset (SSMIS or UMelt) is being used, adjust L107 to "For each station, an 11 x 11 grid of UMelt pixels was centred over each AWS location..."
 • For greater clarity, adjust L112-114 to "These results indicate that around each AWS, the nature of melt conditions is highly homogenous at a scale similar to that of the SSMIS pixel footprint. Therefore, the 6.25 km x 6.25 km resolution of the SSMIS pixel is sufficient to represent local melt conditions and thus is appropriate for calibration purposes."

*We thank the reviewer for these helpful suggestions. We have implemented the proposed wording changes in this section, namely: (i) clarifying that the purpose of the analysis is to assess whether the 6.25 km × 6.25 km resolution of an SSMIS pixel is sufficient to represent melt conditions at each AWS site; (ii) explicitly stating that a grid of UMelt pixels is centred over each AWS location; and (iii) rephrasing the conclusion to emphasise that melt conditions are highly homogeneous at the scale of the SSMIS footprint and that this resolution is appropriate for calibration.*

Second, it is not clear how each UMelt grid was selected, nor how many were analysed through this process. Was this carried out for one grid per AWS or multiple grids per AWS? How was each respective grid chosen? I'm assuming they were only selected for days when melt was observed, but I think it would be good to state this. A short explanation addressing the above questions would be a good addition to this section.

Finally, I am interested to know why an 11 x 11 grid of UMelt pixels was chosen? If my interpretation of your methodology is correct, using a 13 x 13 grid of 500 m x 500 m pixels would produce an overall footprint of 42.25 km2 which: i) is much closer in size to the ~40 km2 footprint of an SSMIS pixel compared to the 30.25 km2 footprint of an 11 x 11 grid, and ii) has a footprint marginally larger than the SSMIS pixel, such that the SSMIS pixel is fully contained within the UMelt grid. Currently, the melt homogeneity evaluated over the 30.25 km2 (11 x 11) UMelt footprint is reflective of only ~75% of the

SSMIS pixel, whereas using a 42.25 km2 UMelt footprint (13 x 13 grid) would enable an evaluation of homogeneity rate and local variability across 100% of the SSMIS footprint. If the homogeneity rate and local variability metrics were computed over a 13 x 13 grid, and produced very similar results, then the conclusion of L112-114 could be strengthened even further, e.g., "These results indicate that around each AWS, the nature of melt conditions is highly homogenous at a scale greater than that

of the SSMIS pixel footprint. Therefore, the 6.25 km x 6.25 km resolution of the SSMIS pixel is sufficient to represent local melt conditions and thus is appropriate for calibration purposes." As the purpose of this section is to evaluate whether the enhanced resolution of the SSMIS pixel is sufficient, even if these metrics decrease, I think it is important to provide a quantification of melt homogeneity over the entire SSMIS pixel footprint.

*Thank you for this helpful comment. We clarified in the manuscript [sec3.1] how the U-Melt grids were selected. For each AWS, a single U-Melt window centred on the station location was used, and melt homogeneity was evaluated for all days in the record (i.e., both melt and non-melt days), rather than only for melt days.*
*Following your suggestion, we repeated the analysis using a 13 × 13 U-Melt window to more closely match the SSMIS pixel footprint. The results changed only marginally (homogeneity >98%, variability <0.02), confirming that melt conditions are highly homogeneous at the scale of a full SSMIS pixel. The manuscript has been updated accordingly.*

3. From my interpretation of the ROC analysis (Figure S1), it appears the two best-performing metrics stated on L132 have been incorrectly identified. Instead of diurnal amplitude, this should be winter anomaly. This error doesn't have subsequent implications for the following stages of analysis, but L132 should be corrected.

*We thank the reviewer for catching this oversight. Indeed, the winter anomaly, rather than the diurnal amplitude, is the second-best performing metric in the ROC analysis (Fig. S2). This was a wording error only and does not affect any subsequent results or interpretation. The sentence on L160 has been corrected accordingly.*

In addition, four of the 'Area Under the Curve' (AUC) statistics shown on Figure S1 are reported as negative, which is not possible (an area cannot be negative). This likely results from plotting the False Positive Rate (FPR) in descending rather than ascending order, causing the integration for AUC to yield a negative value. However, the magnitude of AUC will be correct, and therefore the analysis and interpretation of Figure S1 does not need to be repeated, but the relevant plots should be edited to remove the negative sign.

*Thank you for pointing this out. We have corrected the plotting routine so that all AUC values are now reported as positive. This change affects only the sign of the plotted AUC values; the magnitudes remain unchanged. The updated figure has been included in the revised manuscript (Fig S2)*

In light of the above, I would also suggest reordering the indicators in L141 and in the following lines (L142-147) so that winter anomaly comes before diurnal amplitude.

*Changed accordingly*

**Minor comments**

Throughout the manuscript, the RACMO2.4 dataset used within this study is sometimes referred to as 'RACMO', e.g., L8 or 'RACMO2.4p1' e.g., L197. For continuity and clarity, I would suggest either consistently using 'RACMO2.4' throughout, including within figures, or after the first use of 'RACMO2.4' follow with "hereafter referred to as RACMO".

L30: The acronym 'AWS' has already been established earlier in the manuscript and so "automatic weather station (AWS)" does not need to be written again in full here. Check this here and elsewhere in the manuscript.
*Revised*

L33-34: Would be good here to give spatial resolution and temporal coverage of Banwell et al.'s (2023) dataset.
*Added*

L35-44: The methodology of this study is introduced but there is no mention of the resulting dataset which I think should be included here.
*Revised [L51]*

L50: If AWS data availability is the limiting factor on the temporal coverage of this study's dataset, I think this
point needs to be made more explicitly. Surely these AWSs have data available to the present day?
*Thank you for the helpful observation. We agree that AWS records extend beyond 2021, and AWS availability is not the limiting factor. The actual constraint is the temporal stability of DMSP-F17 SSMIS observations. We have now clarified this in the manuscript [L67/68]*

L51: The manuscript states that the temporal coverage of the dataset is "2012-2021". However, here, the authors note June 2011 – May 2021 as the respective start and end dates of the ten hydrological year-period for which the dataset is produced. Given that the dataset offers annual surface melt rates for the hydrological year 2011-2012 (as shown in first panel of Figure S5), would reporting the temporal coverage of the dataset as "2011-2021" be more inclusive of the actual temporal range? It would also clearly show from a quick glance that it's a decadal dataset.
*We agree with the reviewer and have updated the reported temporal coverage to "2011–2021", which accurately reflects the ten hydrological years included (2011–12 through 2020–21).[L66]*

L56-58: As the SSMIS sensor offers both a standard and an enhanced resolution dataset at 19 GHz frequency, the manuscript should state that the enhanced resolution dataset is being utilised and should use the relevant citation for this (Brodzik et al. (2024): https://nsidc.org/data/nsidc-0630/versions/2)
*Added [L75]*

L57: In the interest of accessibility for all audiences, the manuscript could benefit from a clearer justification for the focused use of the 19 GHz H channel (assuming this was selected due to the general acknowledgement of this channel as having the lowest brightness temp over dry firn?)

*Added [L76]*

L58: As with the 19 GHz channel, the 37 and 91 GHz frequencies also offer standard (25 km) and enhanced resolutions (3.125 km), though it is not clear which resolution is utilised here? It would also be helpful to provide an estimate of penetration depth for each of the three channels. This could provide further context to the authors' statement that utilising the 37 and 91 GHz frequencies didn't notably enhance melt detection; i.e., that most of the detected melt occurs within the sub/near-surface layers of the snowpack. This edit could be addressed at this stage of the manuscript or earlier in the introduction (L16-17).

*We thank the reviewer for this helpful comment. We have clarified in the manuscript which spatial resolution was used for the 37 and 91 GHz channels [L78/80]. As noted, higher-frequency channels are characterised by shallower penetration depths (Colliander et al., 2022), However, we emphasise that we are not able to attribute this outcome with certainty, as we did not perform an in-depth investigation of the physical or instrumental causes beyond the comparative skill assessment.*

L57-59: Horizontal and vertically polarised channels are referred to a lot throughout manuscript and in equations. Introduce the acronyms 'H' and 'V' here and then use consistently throughout.
*Revised*

L66-68: For readers less familiar with AWSs, it would be helpful to give the region in which they are located either in their respective brackets, e.g., "AWS14 (northern Larsen C Ice Shelf, Antarctic Peninsula)" or introduce them by region, e.g., "Four AWSs from the Antarctic Peninsula (AWS14, ...) and three from Dronning Maud Land, East Antarctica (AWS11, AWS16, and Neumayer)...". A useful addition to this manuscript would be a supplementary Figure showing the exact locations of each station. This would give a reader a quick sense of their location and spatial distribution across the Peninsula and Dronning Maud Land.
*A map of AWSs locations has been added in the supplementary document (Fig. S1)*

L73: Adding "into the subsurface layers of the snowpack" here would increase clarity.
*Revised L94*

L76-78: In line with my above comment, give the location of Neumayer station (Ekström Ice Shelf, Dronning Maud Land, East Antarctica) or illustrate its location in a Figure. Here, the authors include Neumayer station with the purpose of broadening the geographic scope for their model calibration to ensure that it performs robustly across climatically distinct regions. Given that four of the AWSs are located on the Peninsula and the other two stations are also located in Dronning Maud Land, East Antarctica, would the inclusion of station data from further afield, such as the West Antarctic Ice Sheet, offer a more comprehensive addition of data to this study? If this is not possible, an explanation of how the inclusion of Neumayer station, in addition to AWS11 and AWS16 (also located in Dronning Maud Land), provides data from a broader geographic scope and climatically distinct region would support these statements.
*We thank the reviewer for raising this point. Unfortunately, no additional AWS sites elsewhere in Antarctica provide the full set of SEB-quality observations required to compute melt consistently with our IMAU AWS stations. This precludes extending the calibration network into other major regions such as the West Antarctic Ice Sheet.*
*Regarding Neumayer station, although its melt regime is indeed similarly low to that of AWS11 and AWS16, the key distinction is data availability: AWS11 and AWS16 provide only a few usable years within our analysis period, whereas Neumayer offers a complete and continuous decade of SEB-quality observations. This longer and uninterrupted*

*record makes Neumayer essential for broadening the calibration beyond the climatology of the Antarctic Peninsula, adding a well-sampled coastal East Antarctic environment to the dataset. We have revised the manuscript to clarify this rationale [L97/101]*

L89-92: I find this section a little confusing and hard to follow. Here are my suggested edits: "To constrain the uncertainty associated with the Then, the uncertainty due to the SEB model, settings and assumptions is estimated by separately varying one each of the five model settings were individually adjusted at the time: i) sensor using a constant height was fixed of the sensors to at 2 m above the surface instead of variable in time, ii) the use of a momentum roughness length was increased from 0.1 mm (typical for snow) to 1 mm for momentum of 1 mm instead of 0.1 mm for snow, iii) using a surface longwave emissivity was decreased from 1 to 0.97 of 0.97 instead of 1, iv) using an alternative snow thermal conductivity was used (Anderson, 1976), and finally v) letting the instead of nudging snow height with sonic height ranger observations, it was allowed to freely evolve in the model instead of prescribing snow height in time using the sonic height ranger observations." For point iv), I think this remains too vague -- what alternative was used? For the earlier list of measurement corrections (L85-87), I would recommend also using the i), ii), iii) … notation, as above, for better readability.
*Thanks for the suggested edits, readability of the section has been improved accordingly.*

L95: Why were sensitivity tests not carried out for all stations? Pointing the reader here to section 3.3 where the sensitivity tests are incorporated and discussed would be beneficial.
*Discussion added [L118/120]*

L102-103 repeats line 105. Remove L102-103.
*Removed*

L117: Is 0.5 mm w.e. day-1 the threshold for identifying a "surface melt day" at the AWSs? If so, I think this could be explicitly stated and justified i.e., why is it not just any day where melt rate is > 0 mm w.e. day-1? Is ± 0.5 mm w.e. the resolution of the AWS measurements?
*Thank you for the comment. Yes, 0.5 mm w.e. day$^{-1}$ is the threshold used to label an AWS day as "surface melt." We added this clarification to the manuscript [L141]. This value was selected to avoid classifying very small, near-zero SEB melt outputs as melt days, since melt is a model-diagnosed quantity rather than a directly measured variable. Near-zero melt values ($\lesssim$0.5 mm w.e. day$^{-1}$) can arise from energy-balance uncertainty, sensor corrections, or rounding in the SEB solution, and sensitivity tests showed that including such values increases noise without altering seasonal melt totals or calibration results. While no community-standard threshold exists for Antarctica, we chose 0.5 mm w.e. day$^{-1}$ that would provide a conservative lower bound that separates physically meaningful melt from numerical noise*

L124: Is there an appropriate reference that could be cited for the Day-to-Day change? Maybe Wang et al.(2025) (https://tc.copernicus.org/articles/10/2589/2016/tc-10-2589-2016.pdf)? Though I appreciate they compare daily Tb values to a previous-3-day average.
*We now cite Wang et al. (2016), who use short-term day-to-day variability in brightness temperature for melt detection, which is conceptually aligned with our "day-to-day change" indicator. The citation has been added accordingly.*

L125: As Abdalati and Steffen (1997) use a cross polarised gradient ratio where both frequency and polarization is evaluated, could a study that uses the normalized polarization ratio be more appropriate? e.g., Mousavi et al. (2021) (https://dspace.mit.edu/handle/1721.1/148558)

*We thank the reviewer for this helpful suggestion. We agree that citing a study explicitly using the normalized polarization ratio is more appropriate in this context.*

L129: Over what time period was this comparison made and, in line with my earlier comment, is AWS-derived melt vs non-melt classified using 0.5 mm w.e. day-1 as an absolute threshold?
*The comparison was performed over the full period of AWS-SEB melt availability used in our calibration (hydrological years 2011–2021). Consistent with the calibration procedure, AWS days were classified as melt when daily melt exceeded 0.5 mm w.e. day $^{-1}$, ensuring that both AWS and SSMIS evaluations use the same melt/non-melt threshold.*

L157: As there are two sets of annual melt-day counts (SSMIS-derived and AWS-derived), for greater clarity be clear which set is being referred to – e.g., "SSMIS-derived annual melt-day counts are obtained by...". Likewise, when referring to "observations" e.g., L81, make it clear these are AWS observations.
*Revised*

For Figure 2, it would be good to report an R2 value for both Figure 2a and 2b to quantify the fit of the data. I suggest labelling each respective y axis as "AWS/RACMO Annual Melt Volume [mm w.e./year]" for clarity.
*Thank you for the helpful suggestion. We have now added the corresponding $R^2$ values for both panels directly in the caption of Figure 2. Axis labels have also been updated for clarity.*

L203-205: It would be nice to visually see these results either as a closer view inset on Figure 3 or as an additional Figure in supplementary materials.
*Thank you for the suggestion. We agree that regional visualisation can be informative, and this is now provided through the annual Antarctic Peninsula melt-flux maps in Fig. S6 of the Supplement. Because these figures already resolve the spatial patterns discussed in Sect. 4.2, adding an additional inset or figure would introduce redundancy without substantially improving interpretation. We therefore opted to retain the current figure layout for clarity and conciseness.*

Here and throughout the results section, be clear about the exact statistics that are being reported, i.e., L204: "melt rates exceeded 350 mm w.e. yr-1... show their highest values on the western inlets..." – are these annual melt rates? decadal mean? highest mean values? For L210 (and elsewhere)."annual surface melt at Roi Baudouin".be clear that these are surface melt rates.
L203-214: Consider presenting your results organised by region (Antarctic Peninsula, West Antarctica, East Antarctica). This will make them easier for the reader to digest. I think it would also be interesting to give regional decadal mean melt fluxes for the Peninsula, WAIS, and EAIS.
L212: Is this region of low-intensity melting a continent-wide minimum for Antarctica or just an area of low melt rates?
*Thank you for the suggestions. We have reorganized the Results section into a clearer regional narrative (Peninsula, West Antarctica, East Antarctica) and clarified the temporal metrics throughout, explicitly distinguishing decadal-mean and annual melt values.*

L224: It is not clear here which products the melt classification is being evaluated for, and therefore what is considered as a false positive?
*We clarified that the misclassification analysis refers to the evaluation of the SSMIS melt classifier against AWS-derived SEB melt [L253]*

L226-231: Surely these discussion points could be reframed with more confidence in the argument presented? "These findings suggest that our classifier is not only responding to surface melt events, but more generally detects the presence of liquid water near the surface..."
The ability of SSMIS to penetrate and detect near/subsurface melt in the snowpack is already acknowledged by the authors early on in the manuscript (L23-24).
"In this sense, SSMIS appears sensitive to a broader melt signal spectrum, including processes not directly measurable by AWS but captured by RACMO's subsurface hydrology." Is SSMIS not already acknowledged for having a broader melt signal given the ability for lower frequency channels to penetrate deeper into the snowpack? It would be interesting to comment on the implications of neglecting shortwave penetration into the snowpack during the AWS SEB modelling (L73).

*We thank the reviewer for this helpful suggestion. We have revised the paragraph to clarify what constitutes a false positive, and we now describe the behaviour of the classifier in a more confident but still cautious way. The revised text emphasises that SSMIS is expected to detect liquid water within the upper firn, consistent with its penetration depth, and that such wetting may not always be diagnosed by AWS-SEB. These clarifications strengthen the interpretation while maintaining physical consistency.[255-263]*

L241: This recent preprint by Zou et al. might be of interest here – https://doi.org/10.21203/rs.3.rs-7384193/v1
L245: Turner et al. (2016) identifies the cooling trend from 1998. I also think it would be worth mentioning the cooling trend was associated with decadal-scale natural variability.
*Revised L281*

L258: Here and elsewhere, the surface melt rate dataset is referred to as "satellite-only/ derived exclusively from SSMIS...". As this study relies not only on satellite-derived brightness temperatures but also AWS observations – as stated by the authors on L46 – I would consider rephrasing from "satellite-only".
*We agree and replaced "satellite-only" with the more accurate wording "SSMIS-derived, AWS-calibrated melt dataset."*

Figure S5: Include in figure caption that these are regional maps of the Antarctic Peninsula.
*Revised*

- *Technical corrections have been all revised.*

*We sincerely thank the reviewer for the detailed, thoughtful, and constructive feedback. The manuscript has improved substantially in clarity, contextual framing, methodological transparency, and interpretive confidence. We believe the revised manuscript now more clearly communicates the novelty and robustness of the work.*

**Response to Reviewer2**

*We thank Reviewer2 for their thoughtful and constructive assessment of the manuscript. Following the editor's advice to change this brief communication into a full paper, we have been able to address the reviewers' comments, including revision and expansion of both introduction and discussions. We appreciate the careful reading and targeted suggestions, which have helped improve the clarity, transparency, and contextual framing of the study. All comments have been addressed in detail below (blue), and corresponding revisions have been implemented in the manuscript.*

Main Points 1. SSMIS data cover the whole of Antarctica, making it possible to generate an Antarctic-wide melt product. However, calibration data are only available for the six AWSs and Neumayer Station. Furthermore, four of the AWSs are located on the Larsen Ice Shelf. The calibration data are thus quite geographically restricted, which begs the question of whether parametrisations derived using these data will be valid across all of the Antarctic melt zone. I think that there are indications that this may be true (e.g., figure S3(c), which, I think, provides model-based evidence that just using these sits for calibration does not introduce major biases) but I'd like to see a bit more discussion of the possible uncertainties resulting from limited calibration data.

*We agree that the limited geographical distribution of AWS calibration data introduces potential uncertainty. We have added a dedicated discussion noting this limitation and clarifying that the strong agreement between the AWS-derived and RACMO-derived melt-day–melt-volume curves suggests good general applicability, while acknowledging remaining uncertainties in poorly sampled regions. The new text has been added in the Discussion (L261–270).*

2. In section 4.2 you discuss the spatial structure of the melt field in the Antarctic Peninsula but it is very difficult to see at the scale of the pan-Antarctic maps presented in figure 3. Given that most melt occurs in the Peninsula region (and that the majority of your calibration data come from this region) I'd consider presenting separate larger-scale maps covering just the Peninsula region.

*Thank you for this suggestion. We agree that regional detail over the Antarctic Peninsula is essential given that most melt occurs there. We note that the Supplementary Material includes annual Peninsula-scale maps (Fig. S5), and we have updated the figure caption to explicitly state that these are regional maps. We now also reference Fig. S5 directly in the Results section to guide readers to the higher-resolution regional view.(L242/243)*

3. Not all readers will be familiar with the locations of the AWSs. I think that a location map would be useful.
*A map of AWSs locations has been added in the supplementary document (Fig. S1)*

*Minor Points (technical suggestions) have been all revised.*

*We are grateful for the reviewer's insightful feedback, which strengthened the manuscript scientifically and editorially. We believe the revisions address all concerns raised. We hope the updated manuscript now meets the reviewer's expectations and we thank them again for their constructive contribution.*